# Ecological Network Analysis for Water Pollution Metabolism in Urban Water Use System: Case Study of Fuzhou, China

**Xiaoping Gao [1,2,3], Yao Zeng [4], Fangying Ji [1,2,*] and Lei Jiang [1,2]**

[1] Key Laboratory of Three Gorges Reservoir Region's Eco-Environment, Ministry of Education, Chongqing University, Chongqing 400045, China; 13805066094@139.com (X.G.); jginger067@gmail.com (L.J.)
[2] College of Environment and Ecology, Chongqing University, Chongqing 400045, China
[3] Fuzhou Planning Design Research Institute, Fuzhou 350108, China
[4] Dadukou District Ecological Environment Monitoring Station, Chongqing 400082, China; hanzhuoo@126.com
[*] Correspondence: jfy@cqu.edu.cn

**Abstract:** Water environment deterioration in urban environments is a critical concern in sustainable water management processes, and the method of urban water metabolism has not been developed more fully in this field. Therefore, there is a requirement to evaluate urban water metabolism with a focus on water quality for sustainable water use. In this study, information and network environ analyses in ecological network analysis (ENA) were explored to measure the water pollutant metabolism state. Six sub-basins in the old part of Fuzhou in China using data from 2016 and 2019 were selected for the case study. Results showed that (1) water pollutant metabolism amount decreased and the metabolism efficiency was improved; (2) the contribution of sub-basins III and IV for pollutant metabolism were more important than the other sub-basins; (3) the river in sub-basin III was the maximum recipient control as a sink node; and (4) ecological relations between compartments were improved for pollutant metabolism. Based on the results, we proposed five types of ENA indicators including TST, $a$, $w_i$, matrix CX, matrix sgnU, and $C$ for the water pollutant metabolism assessment. The method developed here provided new insights to understand the production, transport, degradation, and discharge of pollutants in water use activities in urban environments, and we hope it can be helpful to improve the extension and application of the water metabolism approach in managing urban water quantity and quality in future.

**Keywords:** urban water metabolism; water pollutant metabolism; information analysis; network environ analysis; ecological network analysis

## 1. Introduction

With the rapid development of urbanization, human activity has influenced the urban water cycles greatly. Water shortages and water environment deterioration have become important factors restricting the sustainable development of society, the economy, and the ecological environment in cities [1–4]. In this context, securing water supplies and reducing the impact of socioeconomic water usage on ecological environment require continuous innovations in sustainable water management [5–8].

To understand how efficiently water is being utilized and how the ecological environment is influenced [9], a range of approaches for evaluating urban resource sustainability has been conducted [10–15]. One method that has been studied in recent years is urban water metabolism evaluation [16,17]. This approach focuses on evaluating water flow through urban systems taking different water use processes as production, consumption, and decomposition [9].

The concept of urban water metabolism is developed based on the concept of urban metabolism [18]. According to the definition of urban metabolism, it focuses on the sus-

tainable utilization of material and energy and disposal of waste with a minimum hazard. As the concept is applicable to the hydrologic cycle, which includes water supply, wastewater discharge, and water reuse, Hermanowicz and Asano [16] revisited the concept of urban metabolism from the perspective of the availability of water resources. They then proposed the concept of urban water metabolism with special focus on reclaimed water quality and reuse applications, which described water metabolism in a city with water sources, users and discharges linked with various pathways. Based on the proposed concept and the fact that water shortages are becoming a reality in many cities around the globe, some studies have been conducted to explore the concept and its application in analyzing the characteristics and performance of water metabolism in urban regions (Table 1).

**Table 1.** Chronological review of recent urban water metabolism studies.

| Author (year) | Region of Study | Evaluation Approach | Notes/Contribution |
|---|---|---|---|
| Kennedy et al., 2007 [19] | / | / | Reviewed the previous studies and showed the metabolic processes including water, materials, energy, and nutrient flows that threaten the sustainability of cities. |
| Zhang et al., 2010 [9] | Beijing, China | Ecological network analysis, ENA | Introduced the method of ENA into urban water metabolism research and analyzed the network structure and ecological relationships. |
| Kenway et al., 2011 [10] | Sydney, Melbourne, South East Queensland, and Perth, Australia | Water mass balance, WMB | Formalized a systematic mass-balance framework to quantify all anthropogenic and natural flows into and out of the urban environment. |
| Bodini et al., 2012 [20] | Albareto, Sarmato, and Ravenna, Italy | ENA | Demonstrated the potential of ENA in the urban context through a case study that considered three settlements as the analytic focus. |
| Huang et al., 2013 [21] | / | Material flow analysis, MFA | Conducted a review to highlight the requirement of an integrated assessment of both available and virtual water. |
| Pizzol et al., 2013 [22] | Hillerød, Denmark | ENA | Tested the ENA methodology to determine its generic applicability as a tool for assessing environmental sustainability in urban water management. |
| Renouf and Kenway, 2016 [23] | / | / | Reviewed the utility of existing urban water evaluation approaches to elucidate how they can advance urban water goals. |
| Farooqui et al., 2016 [24] | Ripley Valley Development Area, Australia | WMB | Devised and applied new indicators to evaluate how alternative water source use (storm water/rainwater harvesting, wastewater/gray water recycling) at different scales influences the local water metabolism in a case study on urban development. |
| Serrao-Neumann et al., 2017 [25] | South East Queensland, and the Melbourne and Perth Metropolitan regions, Australia | / | Explored the mechanisms that enable land-use planning to integrate with water resource management based on urban water metabolism concept. |
| Renouf et al., 2018 [26] | South East Queensland, Melbourne, and Perth metropolitan areas in Australia | WMB | Quantified the water performance of urban systems using the water mass-balance framework. |
| Paul et al., 2018 [27] | Bangalore, India | WMB | Tested the urban metabolism framework and analyzed complex urban water systems with real cases in a developing country context. |
| Zhang et al., 2019 [28] | Guangdong Province, China | I-O analysis and ENA | Analyzed the virtual water network situation with the employment of ENA. |
| Zheng et al., 2019 [29] | Guangdong province, China | I-O analysis and ENA | Constructed wastewater metabolism input-output model and analyzed the discharge of industrial wastewater. |
| Liu et al., 2019 [30] | Beijing, China | Emergy analysis | Constructed the urban domestic water supplying process metabolism model and studied it with emergy analysis method. |
| Jeong and Park, 2020 [31] | Seoul, Ulsan, and Jeju, Korea | WMB | Used the urban water metabolism framework to examine patterns of water flows in urbanized areas and evaluated water management performance. |
| Zheng et al., 2020 [32] | Chongqing, China | I-O analysis and ENA | Developed a multi-source virtual water metabolism model to facilitate distributive analysis of the interactive effects from multiple water sources on urban systems. |

By reviewing these studies conducted over the recent years, it can be found that the two main methods that have been developed in parallel are water mass balance (WMB)

and ecological network analysis (ENA). WMB is proposed by Kenway and colleagues [10] and it accounts for all water movement through or stored in a defined volume. By quantifying water flow processes in city-scale and assessing through indicators, this framework has advantages of comprehensiveness and accuracy and can be employed for visioning and for screening improvement opportunities [23]. The other method, ENA, is a complex systems approach applied widely in ecosystem studies [33–36]. Describing water flows as a network of flows between nodes similarly to ecosystems [37], it puts focus on the relationships between components in water flow process rather than the scale of the water flows themselves. ENA is a powerful analytical tool that can help us not only to quantify sustainability but also to reveal component interactions from the perspective of network structure and organizational relationships [36,38]. Therefore, to find a way to give quantitative expression to many of the qualitative observations including how sustainable is water used and how various sectors are connected, ENA is appropriate.

According to the review, we can also find that, despite the increasing number of metabolism studies on water resources, pollutant in water has received relatively less attention. In research of Wu and colleagues [39] conducted in a river basin scale, the various water quality change processes have been investigated and quantified. Their study bridged the gap in water quality metabolism. However, they evaluated water quality metabolism through water volume rather than the amount of pollutant by gray water footprint method. Recently, Zheng and colleagues [29] explored the industrial wastewater metabolism situation to support the industry systems optimization. As the focus of their study is the discharge of industrial wastewater rather than the complete recycling process of all types of sewage, there is still much work need to be executed for modelling and analyzing the total flows of pollutant in urban water metabolism.

Therefore, in order to provide a more comprehensive framework for understanding the characteristics of water pollutant metabolism, with the employment of ENA, this study constructed an ecological network model of urban water pollutant metabolism based on the data of water pollutant in the process of natural and socioeconomic water cycles. The gaps this research fills, and the novel contributions are (1) study urban water metabolism with the focus on water pollutant; (2) a thoroughly network analysis content is carried out including system level and the relationships between system compartments; and (3) the principle for sustainable development of network is investigated regarding the metabolism of hazardous matter.

This paper is organized as follows. Section 2 describes the study site. Section 3 emphasizes the method of ENA, including information analysis and network environ analysis. Section 4 describes the construction of network model. Section 5 presents the information and network environ analyses results and discusses the results in detail. The final section offers several conclusions.

## 2. Study Site

In this study, the old part of Fuzhou city in China was selected as the study site. The study site includes Gulou and Taijiang administrative districts, with an administrative area spanning approximately 30.87 km$^2$ and a total population of 544,000. The water systems in the study area include Meifeng River, Tongpan River, Pingxi River, Fangqin River, Baima River, Luzhuang River, Xinxi River, Daqing River, Wenzao River, Antai River, Dongxi River, Chating River, Xintou River, Jinan River, Zuohai Lake, and West Lake. The total length of the rivers is about 24.5 km. According to the catchment delineation and land use characteristics of each plot, we divided the study area into six sub-basins. The context and boundaries of each sub-basin are shown in Figure 1 and Table 2.

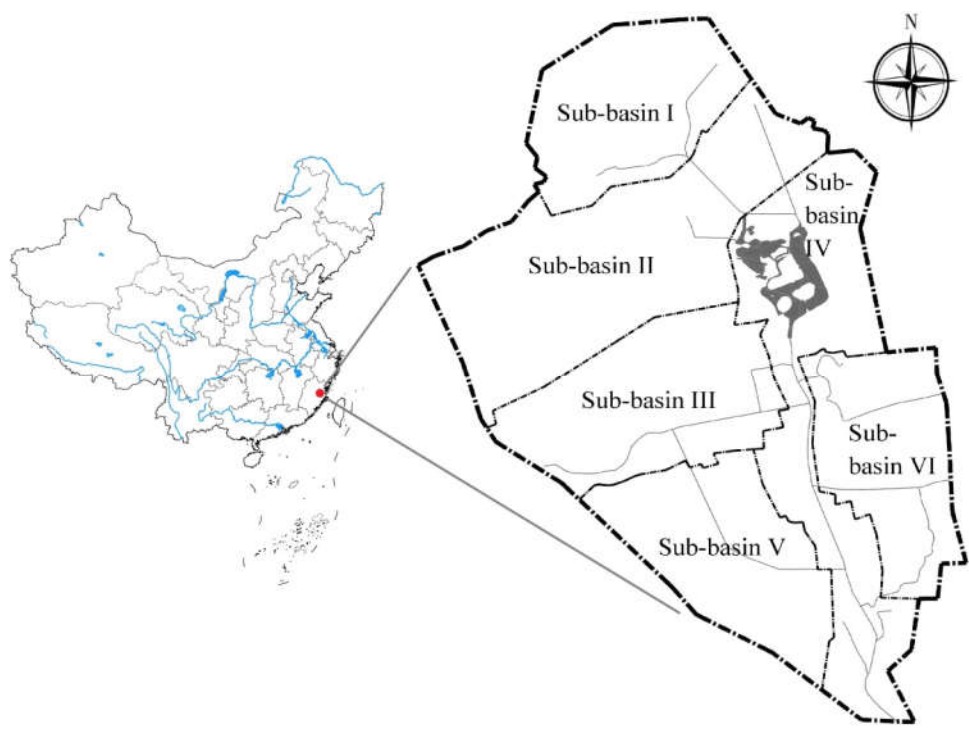

**Figure 1.** Six sub-basins in Fuzhou, China.

**Table 2.** General characteristics of selected six sub-basins in Fuzhou.

| Number | Name | Types of Area | Description |
|---|---|---|---|
| Sub-basin I | Upper reaches of Tongpan River | Residential and industrial area | It covers an area of 2.12 km², primarily including the upper tributaries of Tongpan River. |
| Sub-basin II | Tongpan-Pingxi River | Residential and municipal infrastructure area | It covers an area of 4.56 km², including Meifeng River, Tongpan River, and Pingxi River. |
| Sub-basin III | West Lake-Baima River | Residential, municipal infrastructure, public management and service facilities area | It covers an area of 4.70 km², including Zuohai Lake, West Lake, Fangqinyuan River, Baima River, Xintou River, and Jinan River. |
| Sub-basin IV | Xinxi River | Residential, public management and service facilities area | It covers an area of 4.21 km², including Xinxi River and Luzhuang River. |
| Sub-basin V | Daqing River | Residential, municipal infrastructure and commercial service facilities area | It covers an area of 3.36 km², including Daqing River. |
| Sub-basin VI | Antai-Chating River | Residential, commercial service facilities and municipal infrastructure area | It covers an area of 3.02 km², including Antai River, Wenzao River, Dongxi River, and Chating River. |

Water environmental governance was initiated in six sub-basins in 2016. Through various measures, such as control of pollution source, pipe installment along rivers, sediment dredging, and ecological water replenishment, the water environment quality has been improved, and the impact of point source pollution has been reduced. The residential communities built prior to 2000 have been renovated, and rainwater and sewage diversion work has been initiated. In this manner, the sewage of these communities is included in the municipal sewage pipeline, and the rainwater is included in the municipal rainwater pipeline. In order to reduce the impact of point source pollution on the water environment, the municipal rainwater and sewage pipe networks have been inspected, maintained, renovated and supplemented, and the transport capacities of municipal rainwater and sewage pipe networks have been improved as a result. Additionally, based on the pilot construction of the sponge city, various projects such as greening improvement and ecological slope protection for street parks, beaded parks along the

rivers, and landscapes along roads have been performed. These facilities are beneficial to reduce the harm caused by the non-point source pollution.

## 3. Ecological Network Analysis

ENA is a systematic method widely used in ecosystem research [33–36]. It expresses a complex ecosystem as a system comprising mechanisms for transporting substances, energy, and information between compartments. Based on input-output analysis, ENA focuses on the direct and indirect flows of media between system elements and analyzes the structure, functions, and development and evolution laws of the ecosystem [38]. According to the ecological network modeling theory, if the components of a complex system can be represented as nodes that are associated with each other through the transport and transformation of substances and energy, then this system possesses a specific structure with discrete functions. Such systems can be likened to an ecosystem, thus facilitating modeling and analysis using ENA [37]. Therefore, as an effective method for top-down modeling and analysis, ENA is not only widely used in ecosystem research [40,41] but also applied in socioeconomic system [42], river basin water resource [43–46], urban material and energy metabolism [29,47–49] and urban water metabolism system analysis [9,20,22,28]. In this study, information analysis and network environ analysis in ENA were used to study the metabolic process of urban water pollutants.

### 3.1. Information Analysis

In ENA, MacArthur [50] first applied Shannon's information theory to evaluate the food network in the ecosystem and elucidate the holistic complex characteristics of the network. In information analysis, four indicators, including ascendency (*A*), redundancy (*R*), overhead (*O*), and system development capacity (*C*), are calculated:

$$\text{TST} = \sum_{j=1}^{n} T_j = \sum_{i=1}^{n} T_i \tag{1}$$

$$A = \sum_{i=1}^{n+2} \sum_{j=0}^{n} T_{ij} \log_2 \left( \frac{T_{ij} T_{..}}{T_{i.} T_{.j}} \right) \tag{2}$$

$$\Phi = -\sum_{i,j} T_{ij} \log \left( \frac{T_{ij}^2}{T_{i.} T_{.j}} \right) \tag{3}$$

$$R = -\sum_{i=1}^{n} \sum_{j=1}^{n} T_{ij} \log \left( \frac{T_{ij}^2}{T_{i.} T_{.j}} \right) \tag{4}$$

$$O = -[\sum_{j=1}^{n} T_{0j} \log \left( \frac{T_{0j}^2}{T_{0.} T_{.j}} \right) \sum_{i=1}^{n} T_{i,n+1} \log \left( \frac{T_{i,n+1}^2}{T_{i.} T_{.,n+1}} \right) + \sum_{i=1}^{n} T_{i,n+2} \log \left( \frac{T_{i,n+2}^2}{T_{i.} T_{.,n+2}} \right)] \tag{5}$$

$$C = -\sum_{i,j} T_{ij} \log \left( \frac{T_{ij}}{T_{..}} \right) \tag{6}$$

where $T_{i,j}$ is the amount flowing from compartment $i$ to compartment $j$, $T_i$ is the sum of the amounts flowing into compartment $i$, and $T_j$ is the sum of amounts flowing out of compartment $j$; TST is the total system throughput (Equation (1)).

By calculating the TST, all substance transport and transformation processes, as well as the network scale, can be accurately quantified. The ascendency *A* (Equation (2)) is obtained by calculating the magnitude of the positive feedback in the development process [51,52], and it characterizes the ordered parts that can maintain comprehensive system function for a long time [53]. In comparison to *A*, the indicator *Φ* (Equation (3)) refers to the sum of the system redundancy *R* (Equation (4)) and the overhead *O* (Equation (5)). Through calculating disordered parts of the system, *Φ* characterizes the ability of the

system to recover to the steady state when encountering external interferences, thus indicating system resiliency [54,55]. The sum of $A$ and $\Phi$ constitutes the system development capacity $C$ (Equation (6)). In the study of sustainable development of ecosystems, the balance between $A$ and $\Phi$ is an important feature for the sustainable development of the system [38,56], and it can be characterized using $a = \frac{A}{C}$.

To investigate the water pollutant metabolism network with the application of information analysis: (1) TST is calculated to analyze the total throughput of pollutants in the system, and comparative analyses are then conducted on the changes in total throughput of pollutants in the system before and after water environmental governance; (2) $C$, $A$, $\Phi$, and $a$ are calculated to analyze the transport efficiency of pollutants in the network, and the sustainability of the water pollutant metabolism network can be investigated.

### 3.2. Network Environ Analysis

Based on the traditional Leontief's input-output analysis, Patten introduced the environ concept and proposed a network environ analysis method to study the direct and indirect relationships between system compartments [36]. This method mainly includes flow, control, and utility analyses.

Flow analysis can be used to study the contributions of compartments to the network [57]. Such contributions characterize the position and role of each metabolic flow and compartment in the system. According to the flows between compartments ($f_{ij}$) in the network, matrix N' can be calculated:

$$N' = (n'_{ij}) = (G')^0 + (G')^1 + (G')^2 + (G')^3 + \ldots + (G')^m + \ldots = (I-G')^{-1} \tag{7}$$

where matrix G' represents flows of a given path length, and its element ($g'_{ij} = f_{ij}/T_i$) represents nondimensional, input-oriented flows from compartment $j$ to compartment $i$. The superscript of 0 to $k$ following G' represents the path length. Matrix I is the identity matrix.

Based on the matrix N' and the diagonal of the flow matrix diag(T), the column matrix Y can be obtained through their pre-multiplication:

$$Y = diag(T)\,N' \tag{8}$$

Matrix Y represents the contribution weight of each compartment. By calculating the sum of the elements in each column of matrix Y, $y_j = (y_{1j}, y_{2j}, y_{3j}, \ldots, y_{nj})$, we obtain the total flow ($\sum_{i=1}^{n} y_{ij}$) that compartment $j$ contribute to other compartments. Then, the relative contribution weight can be computed:

$$w_j = \left.\sum_{i=1}^{n} y_{ij} \middle/ \sum_{i=1}^{n}\sum_{j=1}^{n} y_{ij}\right. \tag{9}$$

where $w_j$ represents the importance degree of the compartment $j$ to the other compartments in the systems, which can reflect the role of each compartment in the metabolic process.

By calculating the contribution of each compartment to the other compartment's input and output environs [33,58], the relationship of control between pair-wise compartments in a network can be determined [59]. Similar to the calculation of matrix N', matrix N is calculated based on the matrix G:

$$N = (n_{ij}) = (G)^0 + (G)^1 + (G)^2 + (G)^3 + \ldots + (G)^m + \ldots = (I-G)^{-1} \tag{10}$$

where the element $g_{ij}$ ($g_{ij} = f_{ij}/T_j$) in matrix G measures the dimensionless, output-oriented flows from compartment $j$ to compartment $i$. In order to evaluate the control, a control matrix CX has been defined as CX = $(cx_{ij}) = (n_{ij}/n_{ji})$ [59]. The value of $cx_{ij}$ ranges from 0 to infinity. $cx_{ij} < 1$ indicates that compartment $i$ controls compartment $j$, where $cx_{ij} = 0$ indicates complete control and $cx_{ij} = 1$ indicates no control; $cx_{ij} > 1$ indicates that

compartment *j* has control over compartment *i*, and the control intensity is proportional to the value of $cx_{ij}$.

Utility analysis expresses relationships in interactive networks and is implemented using quantified flow diagrams [36,60]. The direct utility intensity matrix D is calculated:

$$D = (d_{ij}) = (f_{ij} - f_{ji})/T_i \tag{11}$$

where the $d_{ij}$ quantifies the normalized value of net flow between nodes of *i* and *j*. Considering all the indirect influences in the network carried by the higher-order interactions, the sum of all terms of all orders is expressed by the integral utility matrix of U [36,60–62]:

$$U = (u_{ij}) = D^0 + D^1 + D^2 + D^3 + \ldots + \ldots + D^m + \ldots = (I\text{-}D)^{-1} \tag{12}$$

Using the utility sign matrix sgnU, including the gain (+), loss (−), and neutrality (0), the relations between compartments of *i* and *j* can be analyzed. The nine possible binary ecological relationships are shown in Table 3.

**Table 3.** Nine possible binary relationships with the ecological expressions of several generic types [60].

|   | + | 0 | − |
|---|---|---|---|
| **+** | (+, +) mutualism | (+, 0) commensalism | (+, −) exploitation (predation) |
| **0** | (0, +) commensal host | (0, 0) neutralism | (0, −) amensalism |
| **−** | (−, +) exploited (prey) | (−, 0) amensalism | (−, −) competition |

## 4. Construction of Network Model

### 4.1. Construction of Compartments and Paths

According to the transport process of water pollutants in the selected sub-basins and between sub-basins in Fuzhou, the flow of pollutants mainly occurs along five processes (involving both socioeconomic water use and the natural water cycle): domestic water use, industrial water use, sewage purification, surface runoff, and river use. Here, we drew a sketch map to show the process flow diagram for the water pollutant metabolism (Figure 2). From this information and the relationship between the rivers in six sub-basins, we constructed an ecological network model for urban water pollutant metabolism in Fuzhou (Figure 3).

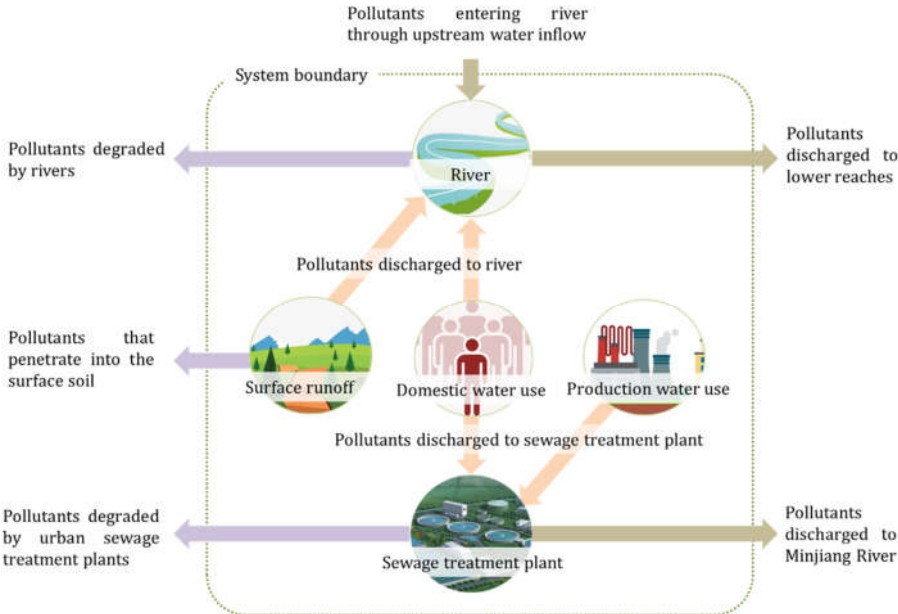

**Figure 2.** Sketch map of the process flow diagram for the water pollutant metabolism in sub-basin in Fuzhou.

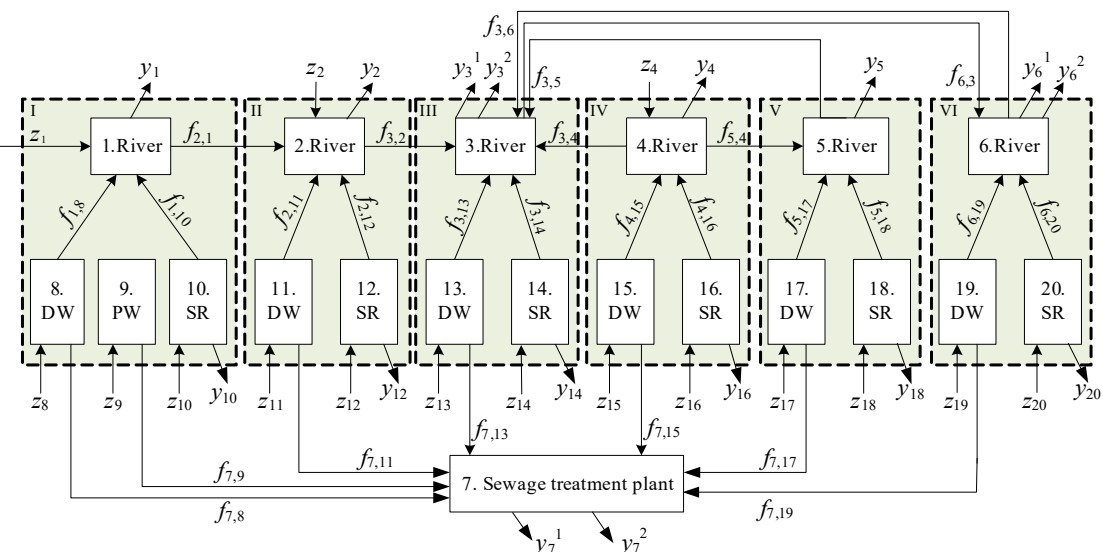

**Figure 3.** Water pollutant metabolism network model for the six sub-basins in Fuzhou.

In Figure 3, $z_i$ represents the number of pollutants imported to the system compartment $i$ from the environment, $y_j$ represents the number of pollutants degraded by compartment $j$ or exported to lower reaches, and $f_{i,j}$ represents the number of pollutants transported from compartment $j$ to compartment $i$. The model contains a total of 20 nodes, and 16 imports, 16 exports, and 26 transport paths between compartments. The descriptions of each compartment and path are presented in Tables 4 and 5.

**Table 4.** Description of water pollutant metabolism network compartments for the study area.

| Compartment | Description |
|---|---|
| River | Surface water bodies in each sub-basin |
| Domestic water use (DW) | Pollutants metabolized in the process of residential domestic water use |
| Production water use (PW) | Pollutants metabolized in the industrial water use process |
| Surface runoff (SR) | Pollutants metabolized in rainfall runoff |
| Sewage treatment plant | Treatment facilities for purifying domestic sewage and production wastewater |

**Table 5.** Description of imports, exports, and flows between compartments in the water pollutant metabolism network model for the study area.

| Type | Compartment | Description |
|---|---|---|
| Imports | $z_1$, $z_2$, $z_4$ | Pollutants entering sub-basins I, II, and IV through upstream water inflow |
| | $z_8$, $z_{11}$, $z_{13}$, $z_{15}$, $z_{17}$, $z_{19}$ | Pollutants produced by human living activities in six sub-basins |
| | $z_9$ | Pollutants from industrial production in sub-basin I |
| | $z_{10}$, $z_{12}$, $z_{14}$, $z_{16}$, $z_{18}$, $z_{20}$ | Pollutants brought by surface runoffs in each sub-basin |
| Exports | $y_1$, $y_2$, $y_3^1$, $y_4$, $y_5$, $y_6^1$ | Pollutants degraded by rivers in six sub-basins |
| | $y_3^2$, $y_6^2$ | Pollutants discharged from sub-basins III and VI to lower reaches |
| | $y_7^1$ | Pollutants discharged to Minjiang River |
| | $y_7^2$ | Pollutants degraded by urban sewage treatment plant |
| | $y_{10}$, $y_{12}$, $y_{14}$, $y_{16}$, $y_{18}$, $y_{20}$ | Pollutants that penetrate into the surface soil during surface runoff |
| | $f_{2,1}$, $f_{3,2}$, $f_{6,3}$, $f_{3,4}$, $f_{5,4}$, $f_{3,5}$, $f_{3,6}$ | Pollutants transported between rivers |
| | $f_{1,8}$, $f_{2,11}$, $f_{3,13}$, $f_{4,15}$, $f_{5,17}$, $f_{6,19}$ | Pollutants in domestic sewage discharged directly into rivers in six sub-basins |
| Flows | $f_{7,8}$, $f_{7,11}$, $f_{7,13}$, $f_{7,15}$, $f_{7,17}$, $f_{7,19}$ | Pollutants in domestic sewage discharged into sewage treatment plants in six sub-basins |
| | $f_{7,9}$ | Pollutants in industrial wastewater discharged into sewage treatment plants in sub-basin I |
| | $f_{1,10}$, $f_{2,12}$, $f_{3,14}$, $f_{4,16}$, $f_{5,18}$, $f_{6,20}$ | Pollutants discharged into rivers through surface runoffs in six sub-basins |

*4.2. Quantification of Network Imports, Exports, and Flows*

4.2.1. Data Source and Processing

(1) Pollution ingredients

In this paper, regarding the data availability, two important pollution ingredients including chemical oxygen demand (COD) and total phosphorus (TP) were chosen as the pollutant metabolized need to be analyzed.

(2) Pollutants transported through rivers

Pollutants are transported from outside the study area to sub-basins I, II, and IV through drainage pump stations, transported between the rivers under the control of sluices, and finally discharged out of the study area through sub-basins III and VI. The pollutants transported in this process, $W_{River}$, can be calculated:

$$W_{\text{River}} = \sum (Q_{\text{River}} \times C_{\text{River}}) \tag{13}$$

where $Q_{River}$ denotes the cross-sectional flow rate, which is taken from the operation data of drainage pump stations and sluices; $C_{River}$ denotes the cross-sectional water quality monitoring data, which was monitored monthly. Here, $Q_{River}$ was calculated as average volume based on two types of operation data in dry and rainy seasons. Water samples for $C_{River}$ were taken at 17 different cross-sections in 11 rivers and 1 diversion tunnel. The 17 sampling cross-sections were shown in Figure S1. The data of $Q_{River}$ were shown in Table S1 and the data of $C_{River}$ were shown in Tables S2-S5.

(3) Pollutants generated and discharged by human living and production activities

The point source pollution caused by human living and production activities in each sub-basin, $W_{Point}$, can be calculated:

$$W_{\text{Point}} = \sum k \times S \times 0.9 \times C_{\text{Point}} \tag{14}$$

where $k$ denotes the highest daily water consumption of different land use types, and its value options can be found in Table S6; $S$ is the plot area of different land use types in a sub-basin; 0.9 is the emission coefficient; and $C_{\text{Point}}$ is the concentration of pollutants in sewage. Because the data of $C_{\text{Point}}$ was incomplete, it is determined here according to the Wastewater Quality Standards for Discharge to Municipal Sewers in China (GBT31962-2015). $C_{\text{Point}}$ of COD and TP were 300 mg/L and 5 mg/L, respectively.

In 2016, the municipal sewage pipe network did not have full coverage. Regarding domestic sewage, some was directly discharged to rivers, while some was discharged into sewage treatment plants through the pipe network before being discharged outside the study area. All industrial sewage was collected in sewage treatment plants for treatment before being discharged. The sewage collection rate of the pipe network in the receiving area of each river before water environmental governance is shown in Table S7. Through pollution interception, the total amount of domestic sewage discharged into rivers in 2019 was reduced by 70%–80%.

(4)   Pollutants discharged and reduced by sewage treatment plants

The effluent of sewage treatment plants complied with the tier-1 class-A standards in the Discharge Standard of Pollutants for Municipal Wastewater Treatment Plants (GB18918-2002), with the concentrations of COD and TP being 50 mg/L and 0.5 mg/L, respectively. Therefore, the reductions of COD and TP by sewage treatment plants were 83.3% and 90%, respectively. The number of pollutants discharged and reduced by sewage treatment plants can be calculated based on COD and TP reduction rates.

(5)   Pollutants flowing from urban rainwater runoffs to rivers

The drainage pipe network in the study area uses a diversion system in which rainwater is discharged into the nearest river according to the area's topography through the pipe network. Therefore, the pollutant load in the runoff is related to the land use type and rainfall levels in the catchment area. According to the average annual rainfall in Fuzhou in the past 30 years and the runoff coefficients of different land use types, the runoff flow in different types of plots, $Q_{\text{Runoff}}$, can be calculated:

$$Q_{\text{Runoff}} = \varphi \times H \times S \times 10 \tag{15}$$

where $\varphi$ is the comprehensive runoff coefficient of a specific land use type; $H$ is the annual rainfall; $S$ is the area of a specific type of plot. The values of the comprehensive runoff coefficient before and after water environmental governance are presented in Table S8.

Runoff pollutants of different surface types during multiple rainfalls were measured to calculate the mean concentrations of various pollutants in the rainwater runoff, *EMC*. The pollutants in the runoff can be calculated as follows:

$$M_{\text{Runoff}} = EMC \times Q_{\text{Runoff}} \tag{16}$$

The load of pollutants flowing to rivers via rainwater runoff in each sub-basin can then be obtained.

4.2.2. Quantification of Network Flows

Based on the constructed ecological network model for water pollutant metabolism in Fuzhou, we collected and processed data to determine the import and export of COD and TP at each compartment, as well as their metabolism between compartments in 2016 and 2019. The results are presented in Table 6. In the table, $z_i$ represents the import of compartment $i$, $y_i$ represents the export of compartment $i$, and $f_{ij}$ represents the flow from compartment $j$ to compartment $i$.

**Table 6.** Ecological network flows of the old part of Fuzhou city in years of 2016 and 2019 (×10$^3$ kg/yr).

| Type | Sign | COD | | TP | |
|---|---|---|---|---|---|
| | | 2016 | 2019 | 2016 | 2019 |
| Imports | $z_1$ | 1567.45 | 1575.56 | 20.49 | 16.83 |
| | $z_2$ | 1720.12 | 1673.49 | 20.87 | 17.55 |
| | $z_4$ | 5845.70 | 5661.60 | 112.72 | 58.26 |
| | $z_8$ | 1217.35 | 1217.35 | 18.42 | 22.57 |
| | $z_9$ | 976.08 | 976.08 | 13.99 | 13.99 |
| | $z_{10}$ | 455.89 | 417.35 | 4.56 | 4.17 |
| | $z_{11}$ | 2533.13 | 2533.13 | 42.22 | 42.22 |
| | $z_{12}$ | 526.47 | 481.98 | 5.26 | 4.82 |
| | $z_{13}$ | 3759.68 | 3759.68 | 62.66 | 62.66 |
| | $z_{14}$ | 690.89 | 632.49 | 6.91 | 6.32 |
| | $z_{15}$ | 3143.75 | 3143.75 | 52.40 | 52.40 |
| | $z_{16}$ | 619.12 | 566.79 | 6.19 | 5.67 |
| | $z_{17}$ | 2650.01 | 2650.01 | 44.17 | 44.17 |
| | $z_{18}$ | 494.12 | 452.36 | 4.94 | 4.52 |
| | $z_{19}$ | 2408.27 | 2408.27 | 40.14 | 40.14 |
| | $z_{20}$ | 444.12 | 406.58 | 4.44 | 4.07 |
| Exports | $y_1$ | 562.69 | 89.48 | 6.76 | 2.89 |
| | $y_2$ | 107.61 | 179.14 | 2.67 | 4.99 |
| | $y_3^1$ | 4704.16 | 472.04 | 8.93 | 23.58 |
| | $y_3^2$ | 7755.88 | 7413.17 | 183.40 | 63.47 |
| | $y_4$ | 125.54 | 52.33 | 4.87 | 6.47 |
| | $y_5$ | 97.51 | 49.24 | 0.36 | 4.40 |
| | $y_6^1$ | 85.38 | 110.39 | −2.75 | 1.33 |
| | $y_6^2$ | 3708.79 | 3809.53 | 68.91 | 27.58 |
| | $y_7^1$ | 9004.04 | 2596.80 | 17.67 | 229.13 |
| | $y_7^2$ | 1844.20 | 12,678.49 | 159.00 | 25.46 |
| | $y_{10}$ | 177.80 | 179.46 | 1.78 | 1.79 |
| | $y_{12}$ | 179.00 | 192.79 | 1.79 | 1.93 |
| | $y_{14}$ | 310.90 | 316.25 | 3.11 | 3.16 |
| | $y_{16}$ | 173.35 | 181.37 | 1.73 | 1.81 |
| | $y_{18}$ | 108.71 | 122.14 | 1.09 | 1.22 |
| | $y_{20}$ | 106.59 | 113.84 | 1.07 | 1.14 |
| Flows | $f_{2,1}$ | 1940.88 | 1855.57 | 27.47 | 18.51 |
| | $f_{3,2}$ | 4919.77 | 3842.89 | 66.13 | 37.36 |
| | $f_{6,3}$ | 5760.55 | 6151.01 | 111.27 | 49.90 |
| | $f_{3,4}$ | 4984.23 | 4367.45 | 93.13 | 40.92 |
| | $f_{5,4}$ | 1967.63 | 1784.43 | 32.28 | 17.34 |
| | $f_{3,5}$ | 3183.04 | 2250.91 | 51.23 | 19.34 |
| | $f_{3,6}$ | 3378.07 | 2846.09 | 66.39 | 29.28 |
| | $f_{1,8}$ | 658.03 | 131.61 | 10.97 | 2.19 |
| | $f_{7,8}$ | 559.32 | 1085.75 | 7.46 | 20.37 |
| | $f_{7,9}$ | 976.08 | 976.08 | 13.99 | 13.99 |
| | $f_{1,10}$ | 278.09 | 237.89 | 2.78 | 2.38 |
| | $f_{2,11}$ | 1018.91 | 203.78 | 16.98 | 3.40 |
| | $f_{7,11}$ | 1514.22 | 2329.35 | 25.24 | 38.82 |
| | $f_{2,12}$ | 347.47 | 289.19 | 3.47 | 2.89 |
| | $f_{3,13}$ | 1375.48 | 412.64 | 22.92 | 6.88 |
| | $f_{7,13}$ | 2384.21 | 3347.04 | 39.74 | 55.78 |
| | $f_{3,14}$ | 380.00 | 316.25 | 3.80 | 3.16 |
| | $f_{4,15}$ | 785.94 | 157.19 | 13.10 | 2.62 |
| | $f_{7,15}$ | 2357.81 | 2986.56 | 39.30 | 49.78 |
| | $f_{4,16}$ | 445.77 | 385.42 | 4.46 | 3.85 |

| | | | | |
|---|---|---|---|---|
| $f_{5,17}$ | 927.50 | 185.50 | 15.46 | 3.09 |
| $f_{7,17}$ | 1722.51 | 2464.51 | 28.71 | 41.08 |
| $f_{5,18}$ | 385.41 | 330.22 | 3.85 | 3.30 |
| $f_{6,19}$ | 1074.16 | 322.25 | 17.90 | 5.37 |
| $f_{7,19}$ | 1334.11 | 2086.02 | 22.24 | 34.77 |
| $f_{6,20}$ | 337.53 | 292.74 | 3.38 | 2.93 |

## 5. Results and Discussion

### 5.1. TST Analysis

Figure 4 shows the TST changes of the water pollutant metabolism network in Fuzhou before and after water environmental governance in 2016 and 2019. The TST values of COD and TP were found to decrease after water environmental governance was initiated. In particular, $TST_{COD}$ decreased from 103,100 tons in 2016 to 98,751 tons in 2019, and $TST_{TP}$ decreased from 1664 tons in 2016 to 1310 tons in 2019. Their network sizes were both reduced.

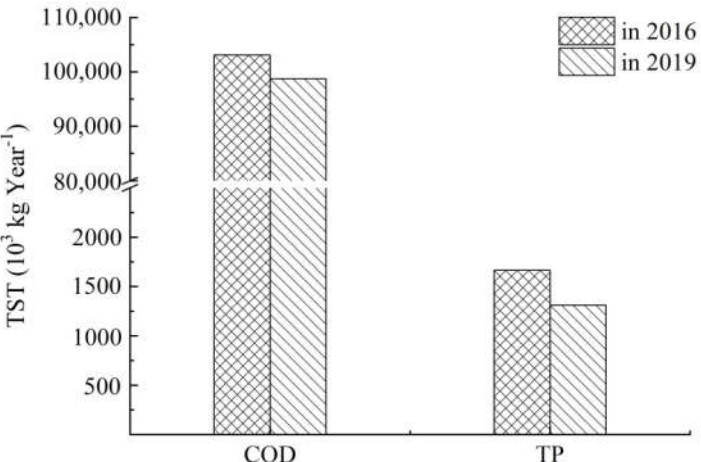

**Figure 4.** TST analysis results in 2016 and 2019 for Fuzhou.

Generally, environmental input to the system and number of compartments and paths are the main factors that influence the network size. In this study, the constructed network compartments neither increased nor decreased before and after water environmental governance; thus, the input from the environment to the system became the main factor influencing changes of the network size. In this study, inputs mainly included three aspects: (1) pollutants entering the sub-basins through upstream water inflow; (2) pollutants produced by living and production activities; and (3) pollutants brought by surface runoffs. Because the six sub-basins did not undergo significant socioeconomic development and changes between 2016 and 2019, and the number of pollutants generated by population changes and production did not change significantly, the changes in the network size were mainly affected by the upstream inflow and surface runoff. Therefore, the reduction of TST indicates that the measures implemented in Fuzhou were effective for decreasing the pollutants in the upstream inflow and surface runoff, from the perspective of the whole system. As TST continued to decrease (i.e., lowered total throughput of system metabolism), water environmental governance became more effective.

In ecology, the expansion of the network size is a main feature of the development of the ecosystem. While, when the flows in network are pollutants, under the premise that

the network compartments and paths remain unchanged, the expansion of the network size may be a manifestation of increased metabolic pressure and system deterioration.

### 5.2. A, Φ, C, and a Analysis

Figure 5 shows the changes in *A*, *Φ*, and *a* of the water pollutant metabolism in Fuzhou before and after water environmental governance implementation in 2016 and 2019. The *A* and *Φ* results indicate that the network development capacity of COD and TP, as well as the ordered and disordered parts of the network, were reduced after measures were implemented. The change in *A* before and after water environmental governance was consistent with the change in TST. This change coincides with the research conclusion of Latham and Scully [63], who report that *A* has a strong correlation with TST, and TST is the dominant factor influencing changes in *A*.

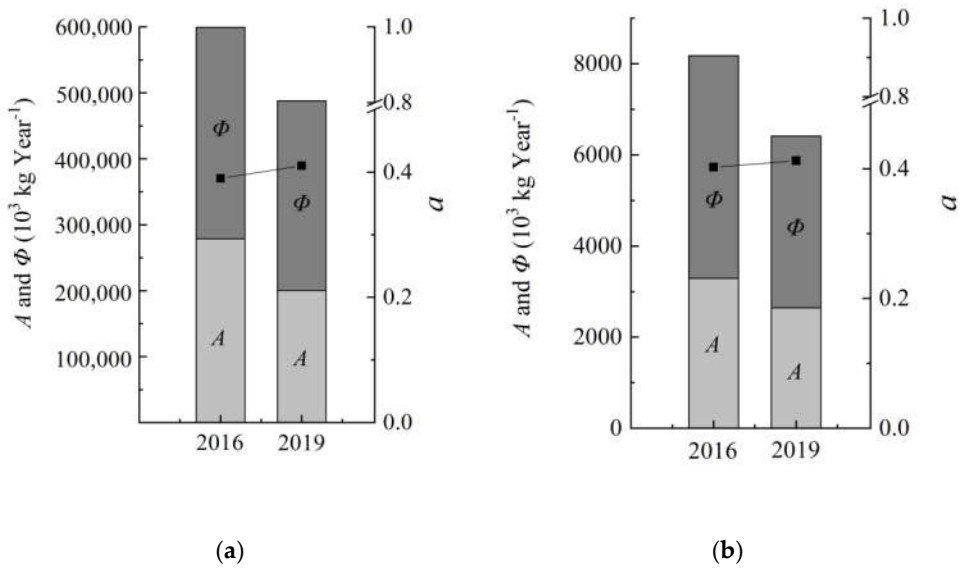

(**a**)                    (**b**)

**Figure 5.** Information analysis results in 2016 and 2019 for Fuzhou: (**a**) chemical oxygen demand (COD) metabolism; (**b**) total phosphorus (TP) metabolism.

The value of *a* increased, indicating that the proportion of the ordered part in the system increased after water environmental governance implemented. According to the ecological network theory, the sustainable development of the ecosystem must maintain a balance between ordered and disordered parts of the system [38,56]. In other words, the network requires a certain number of ordered parts to ensure the efficiency of substance transport in the system [64] and a certain number of disordered parts to ensure that it can recover when subject to external interference [54]. In ecosystem studies, the balance value *a* is 0.4596. However, this theory is not suitable when ENA is applied to the transport and metabolism of pollutants. The relationship between *A* and *Φ* need to be reconsidered.

For water pollutant metabolism, because the process of pollutant transfer is harmful, the less pathways pollutant flows through, the less the influence on environment. Therefore, the criterion for development of pollutant metabolism system should be the simple network pathways. According to ecological network theory, the simple network pathway means the elimination of parallel competing ones, which is expressed by the increase of *a*. In this sense, efficiency of the network structure is beneficial for metabolizing pollutant and the goal function for development of water pollutant metabolism should be the maximum network efficiency. In this study, as the results shown, the *a* values of COD and TP increased from 0.3931 and 0.4023 to 0.4110 and 0.4122, respec-

tively. It indicates that the efficiency of pollutant metabolism has increased with the implementation of water environmental governance in Fuzhou.

To evaluate the performance of water metabolism in studies based on the water mass balance framework, researchers proposed an indicator termed as urban water efficiency, which quantifies total external water use per capita per year [10,23,24,26]. While, to measure the performance of water pollutant metabolism using ENA, we can use $a$ as an indicator, which can provide a systematic evaluation perspective. Using this indicator, for the metabolic performance of water pollutants in Fuzhou, we can therefore determine that the COD and TP metabolism efficiency of the system increased from 0.3931 and 0.4023 respectively to 0.4110 and 0.4122 (by 1.79% and 0.99%, respectively) after water environmental governance. This indicates that the metabolic performance was improved to some extent; however, there remains significant scope for improvement.

*5.3. Flow Analysis*

Tables S9–S16 list the calculation results of the comprehensive flow intensity matrix G′ and contribution weight matrix Y for COD and TP metabolism in 2016 and 2019. On this basis, the weight of the contribution of each compartment to metabolism of water pollutant in 2016 and 2019 was calculated. The calculation results are shown in Figure 6.

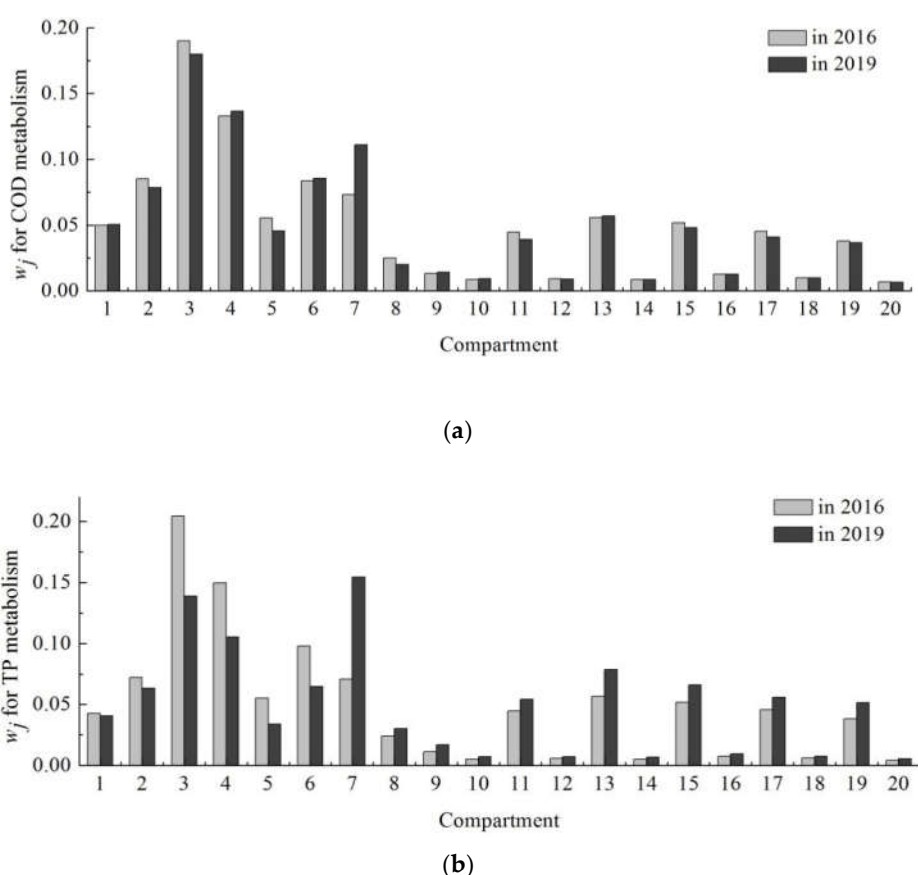

(**a**)

(**b**)

**Figure 6.** Contribution weight of each compartment in 2016 and 2019 in Fuzhou: (**a**) $w_j$ for COD metabolism; (**b**) $w_j$ for TP metabolism.

As shown in Figure 6, the contribution weights of each compartment were different. The weights of most compartments changed slightly in two years except for that of the sewage treatment plant (compartment 7). The weights of the sewage treatment plant for

COD and TP metabolism increased significantly in 2019, which increased from 7.32% and 7.10% to 11.11% and 15.44%, respectively. It is clear that the importance of sewage treatment plant to system metabolism increased. It is related to the implementation of the water environmental governance after which the direct discharge of pollutants into the river was reduced. The improvements in sewage collection and treatment have made the proportion of the pollutants that were degraded, i.e., were metabolized, via the sewage treatment plant increased.

The values of contribution weight of each compartment in two years were averaged and the results are shown in Table 7. The rivers (node 3) in sub-basin III had the highest contribution weight of 15.93%; the sewage treatment plant (node 7) had the second highest contribution weight of 13.28%; the rivers in sub-basin IV (node 4) had the third highest contribution weight of 12.10%; and the contribution weights of other compartments were less than 10%. For the kind of nodes (nodes 10, 12, 14, 18, and 20) which belonged to the pollutant production compartments via surface runoff, their contribution weights were all less than 1%. The ranking result of the contribution weights of the 20 components by node number was as follows: 3 > 7 > 4 > 6 > 2 > 13 > 15 > 17 > 11 > 1 > 19 > 5 > 8 > 9 > 16 > 18 > 10 = 12 > 14 > 20. Therefore, apart from the common sewage treatment plant among the six sub-basins in Fuzhou, the rivers in sub-basins III and IV were more important to the system metabolism.

**Table 7.** Average contribution weight of each compartment for metabolism of COD and TP in 2019 in Fuzhou.

| Compartment | 1 | 2 | 3 | 4 | 5 | 6 | 7 | 8 | 9 | 10 |
|---|---|---|---|---|---|---|---|---|---|---|
| $w_j$ | 0.0456 | 0.0710 | 0.1593 | 0.1210 | 0.0398 | 0.0753 | 0.1328 | 0.0252 | 0.0156 | 0.0082 |
| **Compartment** | **11** | **12** | **13** | **14** | **15** | **16** | **17** | **18** | **19** | **20** |
| $w_j$ | 0.0468 | 0.0082 | 0.0679 | 0.0079 | 0.0571 | 0.0112 | 0.0485 | 0.0087 | 0.0441 | 0.0062 |

For the water transfer relationship in the river network in this case study, it is easy to find that there are more water transport activities between the river in sub-basin III and the rivers in other sub-basins. Therefore, as the key river compartment in the six sub-basins, the rivers in sub-basin III exhibit more pollutant metabolism functions and the metabolism of water pollutants in Fuzhou is relatively greatly dependent on the river in sub-basin III. The compartments that contribute very little to the metabolism of pollutants are mainly pollutant production compartments in surface runoff, which coincides with the current situation that the water environment of Fuzhou is less affected by non-point source pollution than point source pollution.

We classified and summed the contribution weights of 19 compartments other than the sewage treatment plant in 2019 for each sub-basin to obtain contribution weights of the six sub-basins (Figure 7). The figure indicates that the weight ranking result of the six sub-basins was in the following order: III > IV > II > VI > V > I. In particular, the contribution weight of metabolism in sub-basin III was significantly higher than in other sub-basins, with a weight of 23.51%; if the contribution weight of sub-basin IV was added, the total weight of the two sub-basins could reach up to 42.44%. This result shows that the improvement of pollutant metabolism efficiency in the urban water system of Fuzhou should focus on sub-basins III and IV to improve the environmental capacity of the rivers, promote domestic water-saving measures, and increase the treatment rate of domestic sewage.

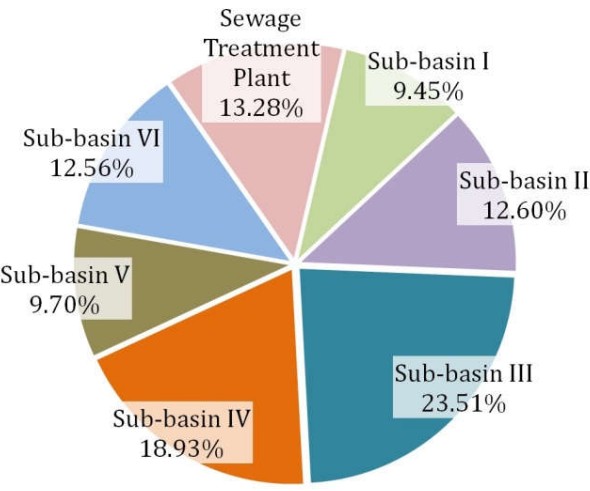

**Figure 7.** Contribution weight of each sub-basin in Fuzhou.

*5.4. Control Analysis*

Tables S17–S20 list the calculation results of the control matrix CX. The results indicate that the control relationship mainly occurred between compartments 1–7 and compartments 1–19. In most cases, compartments 1–7 were controlled by compartments 1–19. To obtain a clearer understanding of the control intensity and dependent objects received by compartments 1–7, we summed the intensity of control received by compartments 1–7 from compartments 1–19. The recipient control values are shown in Figure 8.

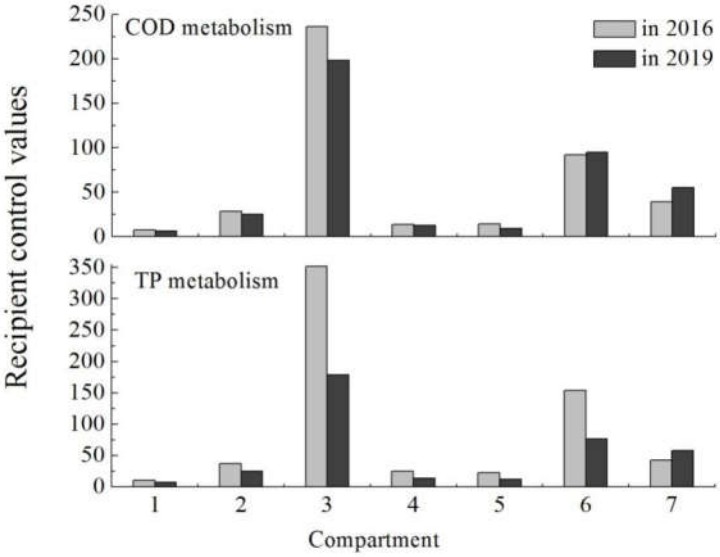

**Figure 8.** Recipient control values for compartments 1 to 7 controlled by compartments 1 to 19.

The results indicate that the rivers in the six sub-basins and the sewage treatment plant were the controlled compartments, among which rivers in sub-basin III received the maximum control and rivers in sub-basin I received the least control. The control intensity was ranked by compartment as follows: 3 > 6 > 7 > 2 > 5 > 4 > 1. According to the control theory in the ecological network [59], the maximum recipient control occurs when a compartment has multiple import paths. At this time, the compartment is under

top-down control. This shows that the river in sub-basin III had multiple import paths in the pollutant metabolism process of the network and served as the pollutant sink among rivers in the six sub-basins. This result shows that the water environment quality of the rivers in sub-basin III as the final sink of pollutants is an important indicator for evaluating the water environment in Fuzhou. The improvement of this indicator can indicate the overall improvement of the regional water environment quality.

By comparing the results obtained for 2016 and 2019, we found that the recipient control intensity of most river compartments was reduced, while that of the sewage treatment plant increased. This indicates that after water environmental governance, the number of pollutants entering the rivers has decreased, and the sewage treatment rate has increased. The distributions of pollutants from the production of pollutants to the import to the rivers and the sewage treatment plant have changed. The role of the rivers as the largest sinks has weakened, whereas the convergence of the sewage treatment plant has been improved. Considering the production, treatment, and discharge processes of pollutants, rivers are the ultimate receptors of pollutants, and their role of sinks will not change. Furthermore, combined with the characteristics of the water system in the river networks in southern China, the maximum recipient control mostly occurs in a specific segment of river.

### 5.5. Utility Analysis

Tables S21–S28 list the calculation results of the comprehensive utility matrix D and matrix sgnU. Tables S29–S32 list the judgment results of the ecological relationship between components.

Fath [60] proposed the division of the number of positive utility by the number of negative utility in the matrix U to obtain a ratio. This ratio is called the mutualism index *M*. The larger the value of *M*, the higher the degree of mutualism. Additionally, Fath [60] has noted that the premise of the application of the mutualism index is that the substance transport between the components in the network should be beneficial rather than harmful. Therefore, in this study, we proposed that the lower the degree of mutualism, the better the system performance. It is because that the water pollutant metabolism networks transport pollutants and the transport of pollutants cannot bring beneficial effects to the network. That is, by direct and indirect interaction between compartments, the less comprehensive and effective utilization between compartments, the less pollutants transmitted in the system, which yields a lower mutualism index and improved system function. Then, according to the meaning of the comprehensive utility matrix and the construction method of the mutualism index, we recommended using the competition index *C*, i.e., the number of negative utility divided by the number of positive utility for the pollutant transport network, to evaluate the overall performance of the metabolism relationship in the network. The value of *C* should be greater than 1, with higher values indicating improved performance. On this basis, the competition index *C* was calculated (Table 8). The results show that the competition index exceeded 1 both before and after water environmental governance executed in Fuzhou, which indicate that the negative utility was always greater than the positive utility among the components and the pollutant metabolism was always maintained in a normal and ordered state.

**Table 8.** Competition index *C* for water pollutant metabolism network of Fuzhou.

|  | 2016 | | 2019 | |
|---|---|---|---|---|
|  | COD | TP | COD | TP |
| Number of negative utility | 214 | 214 | 211 | 209 |
| Number of positive utility | 186 | 186 | 189 | 191 |
| *C* | 1.15 | 1.15 | 1.12 | 1.09 |

To explore the relationships between different types of compartments, firstly, 20 compartments were classified into 5 types: river, domestic water use, production water use, surface runoff, and sewage treatment plant. The classification is as follows: compartments 1 to 6 in the river type, compartments 8, 11, 13, 15, 17, and 19 in the type of domestic water use, compartment 9 in the type of production water use, compartments 10, 12, 14, 16, 18, and 20 in the type of surface runoff, and the sewage treatment plant as compartment 7. Then, based on Tables S29–S32, the number of compartments with competition relationships and the change in the number before and after water environmental governance were calculated. Take the river type as an example. We calculated the number of pairs for each compartment (from compartments 1 to 6) relating with the other compartments competitively, and then the number of pairs was summed. The results are presented in Table 9.

**Table 9.** Number and ratio of the competition ecological relationship for five compartment types.

| Type | Metabolism of COD | | | | Metabolism of TP | | | |
| --- | --- | --- | --- | --- | --- | --- | --- | --- |
| | Number | | Ratio | | Number | | Ratio | |
| | 2016 | 2019 | 2016 | 2019 | 2016 | 2019 | 2016 | 2019 |
| River | 31 | 25 | 25.83% | 20.83% | 27 | 25 | 22.50% | 21.01% |
| Domestic water use | 52 | 55 | 45.83% | 56.25% | 53 | 53 | 44.17% | 44.54% |
| Production water use | 7 | 6 | 5.83% | 5.00% | 6 | 6 | 5.00% | 5.04% |
| Surface runoff | 25 | 28 | 20.83% | 23.33% | 29 | 29 | 24.17% | 24.37% |
| Sewage treatment plant | 5 | 6 | 4.16% | 5.00% | 5 | 6 | 4.16% | 5.04% |

As shown in Table 9, competition relationships mostly occur on compartments for domestic water use, accounting for about 50% of the total. This shows that domestic water usage activities exert a significant restriction on the metabolism of pollutants. As domestic water usage activities primarily include pollutant production and pollutant transport without pollutant degradation, the restriction on compartments for domestic water use is beneficial. Compartments of the types of production water use and sewage treatment plant witness the fewest competition relationships, both accounting for about 5% of the total. For compartment of production water use, the fewer competition relationships was mainly because production water use activities existed only between compartments in sub-basin I and its proportion was relatively low as a result. For the sewage treatment plant, as it mainly provides the function of degrading pollutants, the relatively small proportion of competition relationships on it means that it has little restriction on the degradation of pollutants. Therefore, the ecological relationships in the metabolic system are beneficial to the metabolism of pollutants from the perspective of the proportion of competition relationships in each type of compartment.

The comparison of the results in 2016 and 2019 shows that the proportion of competition relationships in three compartment types (domestic water use, surface runoff, and sewage treatment plant) increased, while that of in river compartment decreased. It suggests that the rivers that have the function of degrading pollutants were less restricted and can degrade pollutants more effectively after water environmental governance, whereas the sewage treatment plant with the pollutant degradation function exhibits greater restriction, and the degradation of pollutants are subject to more competition. In addition, domestic water use and surface runoff compartments that produce and transport pollutants are subject to greater restriction, indicating that their pollutant production and transport functions are inhibited to a greater degree.

The aforementioned changes in the system competition coefficient and competition relationships among various types of compartments indicate several possible conclusions. For the water pollutant metabolism in Fuzhou, the system comprehensively metabolizes pollutants in a normal and ordered state after water environmental governance. Although the competition coefficient has not been improved, the distribution of ecological relationships among the components in the system has been optimized, and the

pollutant metabolism of the system has been improved. In subsequent water environmental governance, it is necessary to further improve domestic and industrial wastewater treatment rates and pollutant degradation rate of the sewage treatment plant.

*5.6. ENA Indicators for Water Pollutant Metabolism Assessment*

To help water management practitioners use this method, we summed the realistic meanings of indicators of ENA in evaluation of water pollutant metabolism in Table 10.

**Table 10.** ENA indicators for evaluation of water pollutant metabolism.

| Indicator | Content | Meaning in practical management |
|---|---|---|
| TST | To assess the total volume of water pollutant metabolism. | The pollutant that produced and discharged should be reduced. |
| *A* | To assess the efficiency of water pollutant metabolism. | The water pollutant transfer nodes and pathways should be simplified. |
| $W_j$ | To assess the importance of compartment in metabolism. | The important object should be identified and be focused preferentially. |
| Matrix CX | To assess the control strength between components. | The pollutant sink should be identified and its water environment quality can be used as the indicative one for the study area. |
| Matrix sgnU and C | To assess the state of relationship between components. | Through scenario analysis, effect of the governance can be assessed from the perspective of metabolism relationship. |

**6. Conclusions**

This study investigated the production, transport, degradation, and discharge processes of pollutants in urban water use activities from the perspective of ecosystem metabolism and based on the ENA method, which compensates for the lack of consideration of water quality in the present studies. The research results show that after implemented a series of water environmental governance projects in 2016, the pressure in terms of pollutant metabolism has been reduced, the metabolism efficiency has been improved, and the ecological relationship between system components has been optimized. In addition, the metabolism of pollutants is in a normal and ordered state. Among the six sub-basins, sub-basins III and IV contribute most to the metabolism of the whole system, and the rivers in sub-basin III serve as the sink of the entire system. This shows that the follow-up water environmental governance work in Fuzhou should focus on sub-basins III and IV and use the water quality of the river in sub-basin III as an indicator for the overall improvement of the water environment in the study area.

In this paper, considering the harmful properties of water pollutants, we discussed the characteristics of the constructed ecological network when using water pollutants as substances flowed in it and clarified the actual significance of the TST change. We also proposed taking *a* as the indicator for evaluating the pollutant metabolism efficiency and revealed the importance of competition relationships in the urban water pollutant metabolism system. Moreover, we analyzed the state of the ecological relationships among the internal components of the system. Because the production, transport, degradation, and discharge of pollutants in urban water use activities are very complex, it is difficult to quantify their data completely and accurately. Therefore, in this paper we combined measured data with empirical formulas to quantify the input, output, and inter-compartment flow of the network to overcome this difficulty. Although the data were not sufficiently accurate and the selection of pollutants was not sufficiently comprehensive, the volume of pollutants transported within and between different users and different sub-basins was in line with the actual situation, which did not introduce errors to the analytic results. In addition, the network compartments in this study area were rela-

tively simple and did not involve many other water use activities, such as the reuse of industrial water, use of agricultural water, and use of reclaimed water in daily life. In future research, the flow data can be made more accurate by increasing the number of measured sites and the frequency of actual measurement. For the abundance of network compartments, representative cities with a more diversity of domestic and production water usage activities can be selected to further analyze the characteristics of the urban water pollutant metabolism process.

In the practice of water management, method of urban water metabolism has not been used and accepted widely. The main reason is the difficulties to quantify the complex natural and socio-economic water cycles and to understand the metabolism meaning. In this study, with the application of ENA, we tried to simplify the modeling of water metabolism process with the consideration of complex direct and indirect relations. In addition, to explore how to guide the actual water management work, the explanation of the real sense that the results indicated was also done. We hope our study can be helpful for the extension and application of water metabolism in management of urban water quantity and quality.

**Supplementary Materials:** The following are available online at www.mdpi.com/2073-4441/13/6/834/s1, Figure S1. 17 sampling cross-sections in Fuzhou., Table S1: $Q_{River}$ in 17 sampling cross-sections [m³/s]., Table S2: $C_{River}$ of COD in 17 sampling cross-sections in 2016 [mg/L]., Table S3: $C_{River}$ of COD in 17 sampling cross-sections in 2019 [mg/L]., Table S4: $C_{River}$ of TP in 17 sampling cross-sections in 2016 [mg/L]., Table S5: $C_{River}$ of TP in 17 sampling cross-sections in 2019 [mg/L]., Table S6: Maximum daily water use for different types of land use [m³/(hm²·d)]., Table S7: Pipeline network collection and discharge rates in 2016 and pollutant reduction rate in 2019 for six sub-basins in Fuzhou., Table S8: Comprehensive runoff coefficient in 2016 and 2019 for six sub-basins in Fuzhou., Table S9: Matrix G' in 2016 for metabolism of COD., Table S10: Matrix G' in 2016 for metabolism of TP., Table S11: Matrix G' in 2019 for metabolism of COD., Table S12: Matrix G' in 2019 for metabolism of TP., Table S13: Matrix Y in 2016 for metabolism of COD., Table S14: Matrix Y in 2016 for metabolism of TP., Table S15: Matrix Y in 2019 for metabolism of COD., Table S16: Matrix Y in 2019 for metabolism of TP., Table S17: Matrix CX in 2016 for metabolism of COD., Table S18: Matrix CX in 2016 for metabolism of TP., Table S19: Matrix CX in 2019 for metabolism of COD., Table S20: Matrix CX in 2019 for metabolism of TP., Table S21: Matrix D in 2016 for metabolism of COD., Table S22: Matrix D in 2016 for metabolism of TP., Table S23: Matrix D in 2019 for metabolism of COD., Table S24: Matrix D in 2019 for metabolism of TP., Table S25: Matrix sgnU in 2016 for metabolism of COD., Table S26: Matrix sgnU in 2016 for metabolism of TP., Table S27: Matrix sgnU in 2019 for metabolism of COD., Table S28: Matrix sgnU in 2019 for metabolism of TP., Table S29: Ecological relationship in 2016 for metabolism of COD., Table S30: Ecological relationship in 2016 for metabolism of TP., Table S31: Ecological relationship in 2019 for metabolism of COD., Table S32: Ecological relationship in 2019 for metabolism of TP.

**Author Contributions:** X.G.: resources, data curation, formal analysis, software, writing—original draft. Y.Z.: methodology, investigation, writing—original draft preparation. F.J.: conceptualization, methodology, project administration, funding acquisition, writing—review and editing. L.J.: investigation, data curation. All authors have read and agreed to the published version of the manuscript.

**Funding:** This work was financially supported by the National Key Research and Development Program (2018YFD1100501).

**Institutional Review Board Statement:** Not applicable.

**Informed Consent Statement:** Not applicable.

**Data Availability Statement:** The data presented in this study are available in Figure S1 and Tables S1-S8.

**Conflicts of Interest:** The authors declare that they have no known competing financial interests or personal relationships that could have appeared to influence the work reported in this paper.

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
