# Peer review of "Ecological Network Analysis for Water Pollution Metabolism in Urban Water Use System: Case Study of Fuzhou, China"

_water, doi:10.3390/w13060834_

Round 1

Reviewer 1 Report

The authors have prepared a comprehensive presentation of the use of the concept of "urban metabolism" to explain changes in water quality in the urban environment. Their literature review and presentation of a new way to include water quality in metabolism studies that have previously focused on water quantity offer to make a worthwhile contribution to the literature.

The paper is in good condition and, in my opinion, can be published in its present form after the usual copy editing job. However, I believe the paper could be improved substantially if the authors would include an assessment of how this information could be used and how.

For example, in the introduction they claimed that the method of using metabolism in urban water studies is well accepted. To the contrary, I believe that while the method has some promise, it is too complex and vague to be accepted by most water management practitioners. This opinion is based on an early interest in the approach, followed by disappointment that it was not embraced by more people and developed more fully. I still think that the method has promise and that the authors are well-positioned to explain this, so I believe that they could increase the value of their paper with this further addition.

Author Response

We're very appreciative of the reviewer’s constructive advice. In the revised manuscript, we revised the original sentence that “One method that has been most commonly used is urban water metabolism evaluation” to “One method that has been studied in recent years is urban water metabolism evaluation”. In addition, we have added two paragraphs in sections 5 and 6.

According to the reviewer’s suggestion, in section 5, we added section 5.6 to sum the indicators of ENA in water pollutant metabolism assessment for a better understanding and utilization in practice. Furthermore, in the final section, we added one paragraph to explain the reasons why the method of water metabolism isn’t well accepted and how we trying to overcome these difficulties through our study.

Reviewer 2 Report

I recommend the publication of this manuscript "Ecological Network Analysis for Water Pollution Metabolism 2 in Urban Water Use System: Case Study of Fuzhou, China". It is well designed and written.

I am only wonder about the calculation of Wriver, could you please explain which Q do you used?Average, maximum?

Also the method of sampling of the river is not clear.

Author Response

Many thanks for reviewer’s comments. We have clarified the calculation of WRiver in the revised manuscript and illustrated the method of sampling with a Figure S1 to show the sampling locations.

Reviewer 3 Report

Please revise the following:

I needed to read the paper to understand the abstract, please revise the abstract include the purpose, methodology and key results. 

The methodology is not clear, can you please include a process flow diagram for the water metabolism,

I believe the pollutant should increase in river due to discharge from sewer treatment plants? there is dilution effect and consumption over the length of the river?

Did you test the sensitivity and accuracy of the ENA?

Any way to justify the results based on scientific discussion?

Please include your answers in the article.

Author Response

1. I needed to read the paper to understand the abstract, please revise the abstract include the purpose, methodology and key results. 

  Considering this reviewer’s recommendation, we have rewritten the abstract.

2. The methodology is not clear, can you please include a process flow diagram for the water metabolism,

   Many thanks for reviewer’s suggestion. In our study, the network model can be considered an illustration of the process flow diagram for the water metabolism. Considering the reviewer’s confusion, as the network model is a little complicated to understand the process, we have drawn a sketch map of the process flow diagram for the water metabolism in the revised manuscript for a clearer description. The added sketch map was shown in Figure 2.

3. I believe the pollutant should increase in river due to discharge from sewer treatment plants? there is dilution effect and consumption over the length of the river?

   In this study, the pollutant coming from the sewage treatment plant discharged to the river of Minjiang. The river of Minjiang is not included in the six sub-basins. To explain it more clearly, we have described the output flows from the sewage treatment plant separately in the added Figure 2 and revised it in the Table 5. There are two types of exports, one is pollutants discharged and reduced by urban sewage treatment plants to Minjiang River, and the other one is pollutants degraded by urban sewage treatment plant.

Yes, there is dilution effect and consumption over the length of the river. However, in our study, as the river was considered as one node in network, the difference along the length of river isn’t included as a result. Also, because the flow used to calculate in ENA is the pollutant amount and which is calculated based on the monitoring data of water quality concentration, the concentration which is correlated closely with dilution effect is not reflected. The effects of dilution and degradation were synthesized and reflected by the pollutant amount in the node of river.

4. Did you test the sensitivity and accuracy of the ENA?

   Many thanks for your constructive question. As a first attempt to model the water pollutant metabolism in urban water use process using ENA, the main aim of this paper is to explore the meanings of the ENA indicator for performance of water pollutant metabolism. Based on this research results, the test of sensitivity and accuracy would be the emphasis in the future work. Through changing the pollutant amount on different pathways and ENA calculations, influences on the results of ENA indicator can be achieved. Then, it would be the important foundation for the proposal of water environment regulatory policy.

5. Any way to justify the results based on scientific discussion?

   We're very appreciative of the reviewer’s constructive comments. In this paper, for the results of two years before and after the water environment measures, our focus was to analyze and discuss the meanings of their changes and their connections with the reality.

   Usually, through scenario analysis method, ENA can play a key functional role in assessment the effect of the governance measures. As for the justification of the results in research of the present state, in this paper, part of the results that can be tested with the reality was discussed. In section of flow analysis, we wrote that “The compartments that contribute very little to the metabolism of pollutants are mainly pollutant production compartments in surface runoff, which coincides with the current situation that the water environment of Fuzhou is less affected by non-point source pollution than point source pollution”. While, for the other results, such as TST, a, matrix CX, matrix sgnU and C, although they are difficult to justify in this study based on the other data because of the attributes of ENA in complex systems, we have tried to explain the rationality of the results.

   In the future work, combining the analyses of sensitivity and accuracy with scenario analysis, and also study the case with many years of data, the results maybe well verified.